# Exploring the Socioeconomic Importance of Antimicrobial Use in the Small-Scale Pig Sector in Vietnam

**DOI:** 10.3390/antibiotics9060299

**Published:** 2020-06-03

**Authors:** Lucy Coyne, Carolyn Benigno, Vo Ngan Giang, Luu Quynh Huong, Wantanee Kalprividh, Pawin Padungtod, Ian Patrick, Pham Thi Ngoc, Jonathan Rushton

**Affiliations:** 1Epidemiology and Population Health, University of Liverpool, Neston CH64 7TE, UK; ianpatrick4229@gmail.com (I.P.); J.Rushton@liverpool.ac.uk (J.R.); 2Food and Agriculture Organization (FAO), Regional Office for Asia and the Pacific, Bangkok 10200, Thailand; carolynbenigno@gmail.com (C.B.); Wantanee.Kalpravidh@fao.org (W.K.); 3FHI 360, Hanoi 100000, Vietnam; VGiang@fhi360.org; 4National Institute of Veterinary Research, Hanoi 100000, Vietnam; lqhuongvet@yahoo.com (L.Q.H.); minhngoc27169@gmail.com (P.T.N.); 5Food and Agriculture Organization (FAO), Country Office for Vietnam, Hanoi 100000, Vietnam; Pawin.Padungtod@fao.org; 6Agricultural and Resource Economic Consulting Services, Armidale 2350, Australia

**Keywords:** pig production, antimicrobial use, antimicrobial resistance, AMR, antibiotic, economics, socioeconomics, antimicrobial use behaviors

## Abstract

Antimicrobial resistance (AMR) is influenced by antimicrobial use in human and animal health. This use exerts selection pressure on pathogen populations with the development of resistance and the exchange of resistance genes. While the exact scale of AMR in Vietnam remains uncertain, recent studies suggest that it is a major issue in both human and animal health. This study explored antimicrobial use behaviors in 36 pig farms in the Nam Dinh Province (North) and the Dong Nai Province (South) of Vietnam (with a median of 5.5 breeding sows and 41 fattening pigs). It also estimated the economic costs and benefits of use for the producer. Data were collected through a structured face-to-face interview with additional productivity data collected by farmers during a six-week period following the initial interview. Overall, antimicrobial use was high across the farms; however, in-feed antimicrobial use is likely to be under-reported due to misleading and imprecise labelling on premixed commercial feeds. An economic analysis found that the cost of antimicrobials was low relative to other farm inputs (~2% of total costs), and that farm profitability was precariously balanced, with high disease and poor prices leading to negative and low profits. Future policies for smallholder farms need to consider farm-level economics and livestock food supply issues when developing further antimicrobial use interventions in the region.

## 1. Introduction

Antimicrobial resistance (AMR) is one of the greatest challenges to global health. The World Health Organization (WHO) have expressed concerns that after 70 years of indiscriminate and overuse of antimicrobials that AMR “threatens the achievements of modern medicine” [1]. Resistant bacteria reside in humans, animals, food and the environment and there are no hurdles to the transmission of resistance genes between these sites or amongst bacterial species. Whilst resistance is a naturally occurring phenomenon, its development is driven through the excessive and inappropriate use of antimicrobials in humans and animals [2,3].

It has been estimated that the economic burden from AMR could be in the region of $100 trillion by 2050, with a worldwide mortality around 10 million and 4.7 million of these deaths occurring on the Asian continent [1,4]. Vietnam has been highlighted as a potential hotspot for the development of AMR due to its high burden of infectious disease, rapidly growing economy and the ease of access to antimicrobials [5]. In response to growing international pressure to tackle AMR, Vietnam launched their first National Action Plan on AMR in 2017 [6,7]. At present, little is known of the burden of AMR on human health and food sustainability in Vietnam with evidence limited to small-scale studies.

In Vietnam, studies show that antimicrobial consumption is high in human medicine with antimicrobials accounting for > 50% of the drugs used [5,8]. An increase in use has been observed in spite of 2005 legislation which requires a prescription to purchase antimicrobials; compliance to this policy has been poor with around 90% of antimicrobials purchased without a valid prescription [8,9]. In parallel, antimicrobial use for animal production is believed to be high in Vietnam, with farmers seeking veterinary advice from pharmacists and obtaining drugs over the counter [10,11].

Studies in Vietnam have identified that antimicrobial use for growth promotion and disease prevention are widespread in the livestock and aquaculture sectors [10,11,12]. Antimicrobials are frequently administered to groups of pigs through medicated feedstuffs [10,13,14] including the widespread use of classes classified by the WHO as being the highest-priority critically important antimicrobials (HP-CIA) for use in human medicine [11,13]. A study by Van Cuong et al. (2016) identified that over half of the commercial pig feed sold through internet retail outlets contained at least one antimicrobial active ingredient. 

In parallel with the high rates of antimicrobial consumption reported, the levels of AMR are believed to be substantial in both humans and animals in Vietnam. AMR presents an increasing threat to public health over time [5,8]. Multi-drug resistant bacteria to Campylobacter, *Escherichia Coli* and *Salmonella* species have been found in both farms and fresh meat samples in Vietnam [11,15,16]. In addition, Kiratisin et al. (2012) found that 43.8% of hospitalized patients in Ho Chi Minh City carried extended-spectrum beta-lactamases [17]. 

Vietnam has a rapidly expanding economy with a Gross Domestic Product (GDP) in the region of $220.4 billion in 2017 and an estimated growth of around 7% in that year [18,19]. Alongside this economic growth, there has been increasing urbanization accompanied by a growing demand for animal source proteins. For example, there has been a 21% increase in pork consumption, from 23 kg per capita in 2006 to 29 kg in 2016 [19]. Despite this transition to a more urban lifestyle the agricultural sector remains the most significant employer in Vietnam, with an estimated 44% of the population employed in the sector. Around 85% of Vietnam’s livestock are raised in small-scale production systems [19,20].

Small-scale commercial pigs units rely heavily on family labor, with an estimated 4 million workers employed in the Vietnamese pig industry [21,22]. Farmers have been under increasing financial pressure since early 2017, with the sector experiencing a turbulent economic scene with falling prices from an oversupply of pork in 2017 [23] resulting in narrow profit margins and pressure on pig producers to maximize productivity. This was further escalated by the emergence of African Swine Fever (ASF) in February 2019, which led to the death or culling of 5.9 million pigs in 2019 [24]. Subsequently, the Vietnamese pig herd has shrunk by around 60% since early 2017, with much of the effect observed in the smaller pig sector [23,24,25]. However, pork prices have stabilized with increasing demand and a depleted supply chain [24]. With mounting international pressure to reduce antimicrobial use in livestock [6], it is essential that countries are able to collect accurate baseline information on current antimicrobial use, explore the behavioral drivers of use, and quantify the economic importance of antimicrobials to key livestock systems. This study addresses a current gap in knowledge and provides baseline information on the role and economic importance of antimicrobials to small-scale commercial pig production and the livelihoods of farmers in Vietnam. The study formed part of a larger framework, which sought to characterize the antimicrobial use/antimicrobial resistance complex in livestock systems in Indonesia, Vietnam, and Thailand [26]. Such research contributes towards developing an evidence base to guide policy-makers in defining feasible regulations to reduce antimicrobial use, whilst maintaining food security.

## 2. Results

### 2.1. Demographic Farm Information

All of the farms included in the study were farrow to finish farms which had either taken part, or agreed to take part, in a previous FAO study into Knowledge attitudes and Practices (KAP) surrounding antimicrobial use which was undertaken in early 2017. Overall, respondent farm sizes were small, with a median of 5.5 sows and 41 fattening pigs across the respondent farms. There were some regional differences observed, with farms in the Southern Province (Dong Nai) generally being larger and more economically important to the household when compared with farms in the Northern Province (Nam Dinh). For example, on the respondent farms on average 100% of the household income was from the pig enterprise in Dong Nai; however, this figure was only 36% in Nam Dinh. The demographic information for the farms is shown in Table 1.

Across the study sample, the main worker on the farm was a manager or owner (80%), with 13% being general farm workers and 8% identified their main responsibility as being a veterinarian. All of the veterinarians worked on farms in the Dong Nai province. Over half of the farms (55%) had two workers or more overseeing the pig business. The average time each worker spent tending to pigs was 35 h per week. This was higher in Dong Nai province where workers spent an average of 48 h per week looking after their pigs, while in Nam Dinh it was 22 h per week.

The amount of time estimated to be spent raising pigs supported previous understanding that pig production is more economically important to households in Dong Nai than in Nam Dinh. In Dong Nai, 89% of all farmers and associated workers regarded pigs as their main occupation, in comparison to 61% of workers in Nam Dinh. Across all the farms the majority of farm workers were male; however, there was a greater number of male workers in Dong Nai province (75%) than compared with Nam Dinh province (50%).

### 2.2. Routine Antimicrobial Use

The routine use of antimicrobial products was reported on all of the respondent farms. Across all of the farms, 82 different antimicrobial products were reported to be used routinely, with 52% of these products being reported in Dong Nai and 48% in Nam Dinh. There was some overlap between the products used in both regions (Figure 1). Overall, 50% of the antimicrobial products contained two or more antimicrobial active ingredients. Therapeutic indications were reported by farmers to be the driver behind 82% of antimicrobial products used routinely across the farms, whilst disease prevention accounted for 11% of products and a combination of treatment and prevention the remaining 7%. 

Injectable formulations were the most frequently recorded products (78%), with in-feed formulations accounting for 21% of products and only one reported use of an oral drench formulation. In-feed antimicrobial use was more commonly reported in Nam Dinh (39% of the antimicrobial products) compared with Dong Nai (5% of the antimicrobial products). However, it is worth noting that the commercially mixed feed observed on some of the study farms did not include a list of ingredients, and as such in-feed antimicrobial use may be under-reported.

Penicillins were the most frequently used antimicrobial class and accounted for 19% (*n* = 116) of the reported antimicrobial usages across all farms, followed by phenicols (16%) and aminoglycosides (16%). The main differences between the Dong Nai and Nam Dinh provinces were in the use of the HP-CIA classes. Whilst the overall use of HP-CIAs only accounted for 20% of the reported antimicrobial usages (*n* = 116) across all the study farms, this figure was only 14% (*n* = 63) in Dong Nai province in comparison to 26% (*n* = 53) in Nam Dinh province. The majority of the HP-CIA use reported across the study farms (83%) was in an injectable formulation, with the other 17% being an in-feed formulation. A breakdown of the reported antimicrobial usages by active ingredient class is shown in Figure 1.

### 2.3. Defining Responsible Antimicrobial Use

Respondents were asked to consider the role of key actors in monitoring responsible antimicrobial use in pigs in Vietnam. The majority (97%) rated the Vietnamese government and veterinarians/para-veterinarians (83%) as being important in monitoring responsible antimicrobial use in pigs (Figure 2). Conversely, only 26.5% of respondents felt that farmers were accountable. The role of market sellers and retailers divided farmer opinion.

The WHO Global Action Plan on AMR advised countries to phase out the use of antimicrobials as growth promoters through assisting with the development and implementation of national AMR policies [27]. Consequently, Vietnam introduced such legislation to prohibit the use of antimicrobial growth promoters (AGP) on 31 December 2017. Overall, the majority of farmers agreed with the AGP ban (86%), with only 8% disagreeing. However, 6% reported that they were unaware of the legislation. A few farmers expressed concerns over the government’s ability and access to resources to enforce the legislation. The majority of respondents (78%) did not think that such a ban would affect the profitability of their farm, whilst 17% thought it would, and 6% did not know. Farmer opinion was divided as to whether the AGP ban would change how they use antimicrobials on their farm, with 56% considering that it would not affect their usage whilst 42% felt that it would and 3% did not know.

### 2.4. The Economics of Antimicrobial Use

All pig farmers (n = 36) reported that disease would reduce profits, 72% of respondents felt that there was an economic advantage in using antimicrobials in their pigs, 25% felt that there was no advantage, whilst 3% did not know. Table 2 identifies some farmer perceptions of the economics of disease and antimicrobial use. 

The initial survey showed that there were differences between the respondents and the demographics of their farms in the two provinces (Table 3). In Dong Nai province the pig populations were larger and both the annual income and the importance of pigs as a source of income was greater in the southern province compared to Nam Dinh in the north. 

Feed cost data were estimated by month and were converted to cost per head. The analysis of these data did not indicate significant differences between the provinces. Farmers found it difficult to differentiate feed between the different ages of pigs and to overcome this problem they were asked to provide an estimate of total purchased feed costs. Whilst not ideal, these were the data available at the farm level and an indication that farm-level monitoring of feed, the major cost of a pig system, was not strong.

The majority of pigs sold were fatteners. In Nam Dinh, the average weight of these pigs when sold was 81.5 kg and the average price was $1.52/kg liveweight for an average price of $122.41 per pig. In Dong Nai, only fattened pigs were sold during this six-week period. The average price for these was $1.23/kg and they averaged 89 kg per pig, for an average value of $109.62. While these prices are realistic for the six-week period of the productivity survey, prices often vary due to various supply and demand factors. Prices are seasonal [28] and are likely to be significantly higher during the Vietnamese New Year celebrations.

From the preliminary consideration of the data from the two provinces, it was decided that there were too many gaps in the data from Nam Dinh province to provide realistic estimates of returns and costs. This is consistent with the understanding that the sector in this province is less commercial than in Dong Nai, with less record-keeping, smaller herd sizes, and potentially fewer commercial pig herds. The following analysis, therefore, was only undertaken for Dong Nai province. 

Producers in Dong Nai indicated that they only sold fatteners during the six-week data collection period. It is unknown whether this is an anomaly or whether farmers do in fact only sell this type of pig. During this period, it was reported that there was a total of 400 fatteners sold at an average price of $109.62. This is an average of 21 fatteners sold per farm in this six-week period or 168.4 fatteners sold per year. These production and price levels were used to estimate income in Scenario 1 of the following Gross Margin (GM) (Table 4). It assumes that prices are not seasonal and, therefore, consistent throughout the year. Scenario 2 alters that assumption by assuming that for part of the year (e.g., Vietnamese New Year) the prices are significantly higher. Scenario 3 uses the annual income estimates elicited directly from the farmers in the questionnaire survey. 

With regard to the costs, only feed, medicines, vaccinations, and routine antimicrobials are included. While there may be other costs (e.g., labor, disinfectants, utilities, equipment) there were not adequate data available on these to include them in the analysis. 

The gross margin analysis indicates that returns have been poor during the time of this survey, with two out of the three scenarios indicating losses to the farmer. This supports the perceptions of the study team and enumerators that the pig sector was facing a difficult economic environment during this time.

The major result, however, is that the medicines and antimicrobials only contribute approximately 2% of costs in this pig production system. As expected, feed is the most significant cost and, therefore, the input that will most concern farmers. 

## 3. Discussion

Internationally, there has been a move towards the intensification of livestock systems to meet growing demands for animal source proteins [29], which generally is associated with larger farm size. Whilst Vietnam has experienced this shift, the majority of the national herd are still housed on small farms with less than 100 pigs [30], with large integrated systems only accounting for around 4% of production in 2014 [31]. All of the farms in the study were considered to fall within this category of small–medium commercial pig herds (< 100 pigs).

The structure of the Vietnamese pig industry implies that pig ownership is common across society and that the pig herd is distributed across a large number of hands. This creates a particular challenge for disseminating information on animal health, communicating responsible antimicrobial use practices, and for the enforcement of policy. These are both financially and logistically difficult for low- and middle-income countries (LMICs). 

Vietnam’s pig industry has experienced a turbulent economic scene since the beginning of 2017, with farmers’ experiencing poor financial returns when the study was conducted in early 2018. Consistently low pig prices, poor profit margins, and an outbreak of ASF have resulted in a contraction in the commercial pig sector since early 2017. The turbulent economic landscape needs to be considered in antimicrobial use policy development to ensure that farms are supported in the journey to reducing antimicrobial use. The measures required to reduce the reliance on routine antimicrobials, such as improved biosecurity, superior management practices and targeted vaccination programs, need to be supported by the government and key livestock stakeholders and be economically viable for producers.

Overall, antimicrobials were a minor component of the farmers’ pig production costs; only around 2% of total expenditure. The low cost suggests that farmers are not under pressure to optimize usage levels and that the use of antimicrobials could well be a factor in minimizing risks, such as productivity losses due to disease, rather than optimizing output and profitability. At present, the potential wider consequences, in terms of negative effects for human health, from antimicrobial use in pigs, are unknown [32]. These factors present a barrier to incentivizing farmers to reduce their reliance on antimicrobials in their production systems. Further work is required to estimate the product development costs, the effects of AMR on future pig production and disease control, and on the present and future effects on human health.

The study results revealed regional contrasts in medicine expenditure, with higher average costs per pig in the southern (Dong Nai) province when compared with the northern (Nam Dinh) province. Similarly, a study into the antimicrobial supply chain for human and veterinary use in Thailand identified regional differences in the availability and costs of antimicrobials [33]. Therefore, the differences in medicine spends observed in the study may relate to regional differences in antimicrobial availability and price rather than differences in antimicrobial usage between the regions. 

A systematic review of global patterns of antimicrobial consumption in pigs identified that use was predominantly for the prevention of disease in the form of an in-feed formulation [34]. In contrast, farmers reported that treatment was the most common reported indication for antimicrobial use and in-feed formulations were seldom reported (< 20% of all uses). In addition, researchers noted that purchased feed frequently did not include a list of ingredients, a circumstance observed in other studies into antimicrobial use in pigs in Vietnam [11,13]. Therefore, in-feed antimicrobial use is likely to be under-reported. Concerns over misleading and imprecise labelling of commercial livestock feeds has been identified as a priority area for AMR policy in South East Asia [35].

The use of combinations of more than one antimicrobial active ingredient was commonplace in the study and is a behavior identified in other published research in the Vietnamese pig sector [13,14]. This behavior is in opposition to international guidelines on antimicrobial stewardship which recommend the use of the most appropriate antimicrobial for the causative pathogen, and that where possible this should have a narrow spectrum of activity [36,37,38]. There is scope for further educational initiatives to educate farmers and other key stakeholders on the principles of responsible antimicrobial use.

There has been significant international pressure to discontinue the use of HP-CIAs in livestock species over concerns of the potential effects on human health [39,40]. However, in spite of this pressure their use remains widespread in pig production both in Vietnam and on a global scale [11,13,14,34]. In agreement, the study showed that HP-CIA classes accounted for 20% of routine antimicrobial use on the study farms. 

In parallel with many other LMICs, there was no requirement for a prescription for antimicrobials for use in either human or veterinary medicine in Vietnam at the time of study [7]. Antimicrobials for use in livestock are freely available through feed companies, pharmacies, drug sellers, and directly from pharmaceutical companies [10,14]. Evidence from human pharmacies highlights poor knowledge of prudent antimicrobial use by drug sellers, inappropriate use practices, and the easy availability of HP-CIA classes [9]. Thus, it seems likely that advice farmers receive from pharmacies and drug sellers may be diverse, conflict with responsible antimicrobial stewardship guidelines, and may facilitate indiscriminate antimicrobial use.

In compliance with the international adoption of the WHO global action plan on AMR, Vietnam has introduced a ban on the use of AGPs [6]. Overall, case study respondents agreed with the 2017 ban; however, 6% of respondents reported that they were unaware of the legislation. More recently, the Vietnamese government has taken a stepwise approach to prohibiting the use of all antimicrobials for prophylaxis by the end of 2025, with an initial ban of the use of the HP-CIA classes by the end of 2020 [41]. However, a grey area exists between the use of antimicrobials for prophylaxis and growth promotional indications, as both may warrant the routine, and long-term, use of sub-therapeutic doses of antimicrobials [42,43]. This highlights a need for further knowledge exchange with producers, veterinarians, and key livestock stakeholders with regards to the definition of AGPs and prophylaxis and what practices are prohibited by the legislation. 

Some farmers in the study also expressed concern that the Vietnamese government may lack the resources to properly implement and enforce the AGP ban. Therefore, in order to ensure compliance any future regulations on antimicrobial use would require significant resources and capacity which at present are severely lacking [44]. This is an area highlighted as a concern throughout South East Asia which requires enhanced financial support and resources to properly enforce the legislation [35].

AGPs are considered to be particularly beneficial for disguising subclinical disease on unhygienic and older livestock housing systems [45]. Consequently, there are concerns that the AGP ban will have wide-reaching negative economic effects on Vietnamese pig production through uncovering endemic disease signs in herds. The majority of respondents considered that antimicrobial use offered an economic advantage for pig businesses. Therefore, it is essential that producers receive sufficient financial support to seek alternative management strategies for preventing disease. Without sufficient investment it is likely that routine antimicrobial use will continue to be the norm.

The national action plan on AMR proposes a future requirement for prescription for the use of antimicrobials in animals; however, introducing and enforcing such legislation presents multiple challenges due to competing agendas and conflicts of interest of private sector organizations. Going forward, a stepwise approach to limiting antimicrobial availability would be a sensible route to achieving the ultimate goal of prescription-only antimicrobial use in veterinary settings. This process has started already started with the 2017 ban on AGP [7].

## 4. Materials and Methods

### 4.1. Study Design and Setting

The study was part of a larger Food and Agriculture Organization (FAO) funded project to create a framework to characterize the antimicrobial use/antimicrobial resistance complex in livestock in Indonesia, Thailand, and Vietnam [26]. It was undertaken through a collaboration with the FAO country office for Vietnam and the National Institute for Veterinary Research (NIVR). Farms were recruited as a convenience sample selected by the FAO and were split between Dong Nai Province (n = 19), located in the Southern region of Vietnam, and Nam Dinh Province (n = 17), located in the Northern region. The sample farms had either taken part, or agreed to take part, in a previous FAO study into Knowledge attitudes and Practices (KAP) surrounding antimicrobial use which was undertaken in early 2017. The farm sample encompassed small (<19 breeding sows) and medium (20–49 breeding sows) commercial pig farms in the regions. All farms were easily accessible by road. 

### 4.2. Data Collection

The research team visited each farm twice in early 2018. On the first visit in January 2018, the NIVR research team conducted a face-to-face structured interview using a detailed questionnaire. Farmers were then requested to record productivity, animal health, and antimicrobial use data over a six-week period as a productivity survey. Provincial government veterinary officers visited each farm after weeks two and four to assist with capturing these data. The productivity survey data were collected on a second farm visit by NIVR researchers in March 2018. Data were entered in Vietnamese and translated into English by researchers at NIVR.

### 4.3. Questionnaire Design

The questionnaire and productivity survey were developed by researchers at the University of Liverpool with guidance from the FAO and NIVR to ensure that the content was relevant to the Vietnamese pig sector. The questionnaire content was complimentary to the KAP study and covered content and subject areas not previously captured. The questionnaire was developed in English and was translated into Vietnamese by researchers at NIVR. 

The questionnaire consisted of the following sections:
Demographic farm and respondent information.Demographic information on the type of pig production, the workers employed on the farm, the herd size, the economic importance of pigs for the farm, the location of the farm, feeding practices, management practices, and productivity data on the pig production enterprise.Antimicrobial use on farm.Information on antimicrobials routinely used on the pig farm; formulation, active ingredient, pack size, the disease indication for use, whether use was preventative or therapeutic, course and duration. For the purposes of the study, the WHO definition of the high-priority critically importance antibiotics (HP-CIAs) was adopted which refers to the macrolides, fluoroquinolones, third and fourth generation cephalosporins and polymixins (colistin) with regards to use in livestock.Attitudes to responsible antimicrobial use.Attitudes to the responsibility of antimicrobial use, knowledge and awareness of antimicrobial use policy.The economics of antimicrobial use.The economic questions explored the profitability of the pig enterprise, antimicrobial costs, feed costs, prices obtained for selling pigs, average bodyweight at slaughter, weight at point of sale, mortality rates and other medicine management costs.

### 4.4. Questionnaire Data Analysis

Data analysis was undertaken using Microsoft Excel 2016 (Microsoft Corporation, Redmond, WA, USA) and SPSS Statistics 24 (IBM SPSS Statistics for Windows Version 22.0; IBM Corp., Armonk, NY, USA). Descriptive statistics were presented as percentages of response categories or Likert-scale responses. Responses for open text box questions were coded into categories and presented as a percentage response; these responses were supported with relevant quotations, which represented majority opinions. All results were presented both as a whole sample and also at a provincial level. 

### 4.5. Economic Analysis

The productivity survey captured detailed information on the economics of pig production on each study farm for a six-week period. This included collecting data on feed costs, routine antimicrobial use costs, other medicine costs, number of pigs sold and prices obtained for pigs sold. 

Medicine costs were collected on a per cycle basis. A cycle generally related to the 3.5 month fattening period. Fattening would begin with a piglet weighing about 7 kg (22 to 28 days old) and end with a fattened pig being sold at 100 kg. For a farm with breeding sows and fattening pigs, the cycle from farrowing to selling the finished pig would be between 4.5 to 5 months. This analysis, therefore, assumes 2.5 cycles per year with approximately 3 months of that period being the period of managing piglets and the remaining 9 months being fattening pigs. Gross Margin analysis assumes that prices are not seasonal and, therefore, consistent throughout the year.

The analysis generated valuable gross margin (gross income less variable costs) information per farm and per fattened pig and evaluated the importance of medicine costs in decision making. The production and price variables were varied in order to examine the sensitivity of the financial results to changes in market conditions and production performance.

### 4.6. Ethical Approval

Ethical approval was granted by the University of Liverpool Veterinary Science Research Ethics Committee which also required proof of ethical acceptability in Vietnam (Reference 635 number VREC640). As the study did not involve the collection of samples from animals or humans, the research collaborators and local government in Vietnam did not require a specific ethical review. Therefore, NIVR provided documentation mitigating the need for a detailed ethical review in Vietnam. 

## 5. Conclusions

There are a large number of antimicrobials used in the livestock sector in Vietnam, and AMR may be widely prevalent [46]. Weak or non-existent regulatory frameworks governing antimicrobial use, sub-optimal enforcement and compliance with existing guidelines, low levels of AMR awareness, and inadequate commitment to responsible antimicrobial use stewardship are driving the development of AMR. This weak scenario will not have been helped by the ASF outbreak and will put further difficulties on a transition to the responsible use of antimicrobials. However, the national action plan on AMR recognizes that reducing antimicrobial use in livestock needs a new approach to animal management which encompasses improving husbandry practices, superior biosecurity, and more targeted use of vaccinations. Thus, whilst significant changes in antimicrobial use practices are likely to be a future aspiration, there is the understanding and motivation to tackle the issue at a national level. 

## Figures and Tables

**Figure 1 antibiotics-09-00299-f001:**
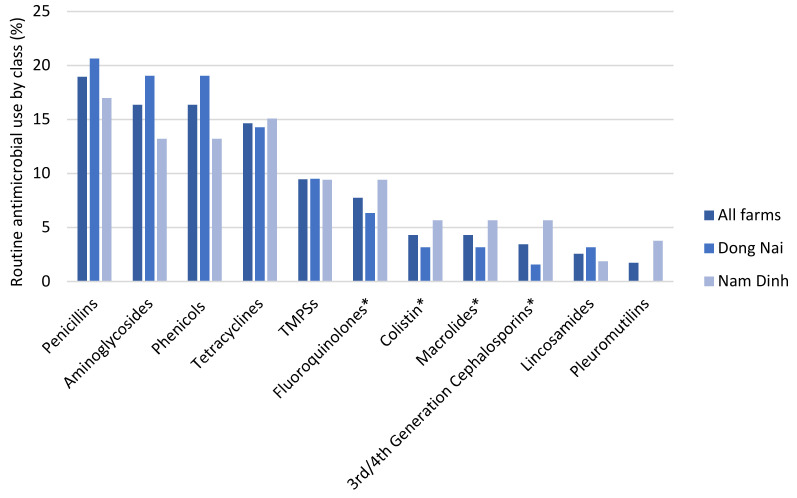
Routine antimicrobial use by antimicrobial class (%) (n = 116). In the case of a combination antimicrobial product, each active ingredient is recorded separately. * = Antimicrobials identified by the WHO as HP-CIAs; *TMPSs—Trimethoprim sulfonamides*.

**Figure 2 antibiotics-09-00299-f002:**
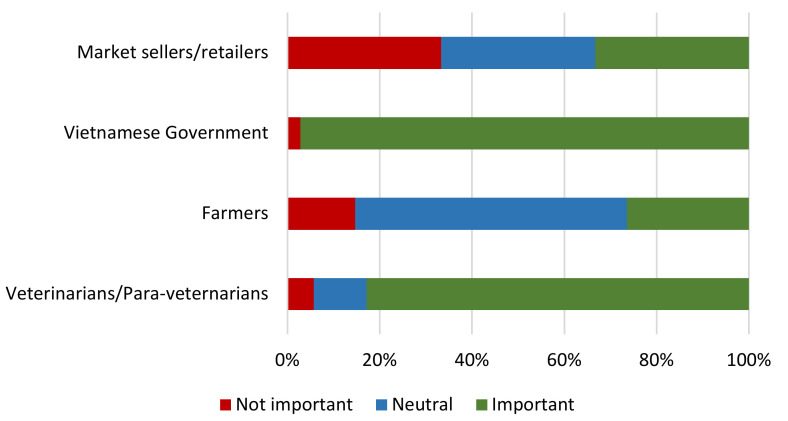
Farmer perceptions of the role of different actors in monitoring the use of antimicrobials in pigs.

**Table 1 antibiotics-09-00299-t001:** Demographic information for the farms in the study into antimicrobial use in commercial pig production in Vietnam.

		All Farms (n = 36)	Dong Nai (Southern) (n = 19)	Nam Dinh (Northern) (n = 17)
**Information on farm size and productivity data recording**				
**Median number of pigs on farm at time of survey (Minimum–maximum, IQ range *)**		Sows 5.5 (1–40, 7.5) Piglets 20 (7–80, 39) Fatteners 41 (1–250, 76.25)	Sows 10 (1–40, 20) Piglets 45 (10–80, 30) Fatteners 85 (10–250, 125)	Sows 4 (1–11, 4) Piglets 17 (7–50, 17.25) Fatteners 20 (1–60, 31.5)
**Proportion of farms with productivity data recorded**	**Number of pigs**	75%	100%	47%
	**Health status**	67%	84%	47%
	**Profit**	28%	42%	12%
**Information on farm workers**				
**Number of farm workers across the farms**		56	28	28
**Proportion of farm workers where pigs are their main occupation**		75%	89%	61%
**Gender**	**Female**	37%	25%	50%
	**Male**	63%	75%	50%
**Main role**	**Farm manager/owner**	80%	82%	79%
	**Veterinarian**	7%	14%	0%
	**Other farm workers**	13%	4%	21%
**Median number of hours each worker spent with pigs (Minimum–Maximum, IQ range *)**		35 (3–56, 35)	48 (21–56, 19.25)	22 (3–56, 16.5)

* IQ range—interquartile range.

**Table 2 antibiotics-09-00299-t002:** Farmer perceptions of the economics of disease and antimicrobial use.

**Farmers shared the opinion that disease problems had a negative effect on the profitability of the farm**	***“We must pay for treatment; pig grow slowly, and can transmit diseases to healthy pigs.”*** ***“It causes losses of pigs, reduced weight gain and then causes economic losses for the family.”*** ***“We have to pay for medicines, effects on productivity, we have to feed longer.”***
**Farmer perceptions of whether there is an economic advantage to the use of antimicrobials in pigs**	**Economic advantage to antimicrobial use in pigs:** *“Pigs are healthy after treatment so I can sell it as planned.”* *“We can prevent diseases, pigs can grow quickly, farmer feels able to do their job.”* *“Yes, because it decreases outbreak of diseases.”* **No economic advantage of antimicrobial use in pigs:** “*Good prevention makes healthy pigs, so that saves money. When pigs are ill, we must spend money for treatment, so that we lose money.”* “*No. We have to pay for antibiotics for treatment.”*

**Table 3 antibiotics-09-00299-t003:** Pig production and farm income data (min and max in brackets) (USD) *.

	All Farms (*n* = 36)	Dong Nai (Southern) (*n* = 19)	Nam Dinh (Northern) (*n* = 17)
**Average number of pigs sold per month**	18.3 pigs (2–60)	25.8 pigs (2–60)	9.9 pigs (2–50)
**Average proportion of annual income from pig production**	63.3% (20–100%)	74.7% (20–100%)	50.6% (30–100%)
**Average annual income from pigs ($)**	16,000 (0–64,000)	28,500 (2100–64,000)	2000 (0–6900)
**Average cost of medicines per pig per cycle ($)**			
**Adult pigs**	5.91	6.64	3.14
**Piglets**	4.89	5.15	4.17
**Fatteners**	6.12	6.64	5.29
**Average feed cost per month ($)**	1355	2100	350

* $1 = VND23300.

**Table 4 antibiotics-09-00299-t004:** Pig production and farm income data (minimum and maximum in brackets) ($) * Annual Gross Margin for pig enterprise in Dong Nai.

Scenario	Item	Unit	Number	Price Per Unit ($)	Total Per ($)	
					Farm	Fattened Pig
**Income**						
**Scenario 1:**	**Fattener price—Prod survey**	**Pig**	168.40	109.62	**18,462**	**109.63**
**Scenario 2:**	**High price fatteners**	**Pig**	33.70	191.35	**21,214**	**125.97**
	**Low price fatteners**	**Pig**	134.70	109.62		
**Scenario 3:**	**Fattener price—Survey 1**	**Unit**	1	28,507	**28,507**	**169.28**
**Costs**						
**All scenarios treated the same**	**Feed**	**Unit**	1	25,176	25,176	149.50
	**Medicine**	**Unit**	1	55	55	0.33
	**Routine AMs**	**Unit**	1	414	414	2.46
				**Total Cost**	**25,645**	**152.29**
**Gross Margin**						
**Scenario 1:**					**−7183**	**−42.65**
**Scenario 2:**					**−4431**	**−26.31**
**Scenario 3:**					**2862**	**17.00**

* $1 = VND23300

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
