# Peer review of "Exploring the Socioeconomic Importance of Antimicrobial Use in the Small-Scale Pig Sector in Vietnam"

_antibiotics, 2020, doi:10.3390/antibiotics9060299_

Round 1

Reviewer 1 Report

The manuscript by Coyne and coworkers is well-written and fits well within the framework of the special section "Antibiotics Use and Antimicrobial Stewardship".

The article by the same research group: Antibiotics 2019, 8, 33; doi:10.3390/antibiotics8010033 should be cited and similarities between that and the present manuscript clearly stated

Author Response

Many thanks for your supportive comments. The study formed part of the wider project to develop a framework to characterise the antimicrobial use/antimicrobial resistance complex in livestock in Indonesia, Thailand and Vietnam. The authors want to thank you for identifying that this valuable background information is missing from the manuscript. In response, a sentence to provide context to the study has been added to the introduction (lines 93-96) and the material and methods sections (lines 335-338). These additions also include a reference to the paper Coyne et al. (2019) referred to.

Reviewer 2 Report

There is a lack of novelty in this manuscript. Data included do not demonstrate the objective; these only show costs and the results are weak.

The manuscript should be highly improved.

Antibiotics-806666
Exploring the socioeconomic importance of antimicrobial use in the small-scale pig sector in
Vietnam
There is a lack of novelty in this manuscript. Data included do not demonstrate the objective;
these only show costs and the results are weak.
ABSTRACT
Line 22. It says 36 pig farms. Please, give a mean size: number of sows, finishers…
Lines 28-30. There are no data at all. Data are necessary.
KEYWORDS
The following keywords are not adequate: AMR, Vietnam and behaviours.
INTRODUCTION
Line 71. What is GDP?
Lines 86-88. Objectives are not adequately expounded.
RESULTS
Line 92. More information about farms is missed. Please, give standard deviation; at least in
table 1.
Table 1. Is “number of farm workers” considering all the farms?
Line 113. Routine antimicrobial use. Does “use” refer to doses, animal or amount of
antimicrobial active product? It is not clear.
Lines 161-166. This not enough to study “Farmer perceptions on the economics of disease and
antimicrobial use”.
Table 2. Delete it, because this is for a journal and no a newspaper.
Table 3. These results are very light.
Lines 180-185. These are material and methods.
Lines 202-209. These are material and methods.
DISCUSSION
Lines 233-247. This is stuffing text. Please, reduce it or delete it.
Lines 311-313. These are good conclusions.
MATERIALS AND METHODS
Lines 322-323. These are objectives; it is not methodology.
Lines 323. It says “small-medium commercial pig sector”. How are the farms? Number and kind
of animals?
Lines 360-363. The economics of antimicrobial use is not compared. There is not information
beyond what is spent; who do you know the antimicrobial effect?
CONCLUSIONS
The conclusions are very weak; these were well known statements.

Author Response

The authors would like to thank the reviewer for taking the time to provide valuable insight and comments on the manuscript. The authors have provided a detailed response below in Italics to the reviewer comments.

There is a lack of novelty in this manuscript. Data included do not demonstrate the objective;
these only show costs and the results are weak.

Many thanks for taking the time to provide comments.

ABSTRACT
Line 22. It says 36 pig farms. Please, give a mean size: number of sows, finishers…

The median number of breeding sows and fattening pigs has been added in the abstract (line 23). The average was shown by the median as these data were not normally distributed.

Lines 28-30. There are no data at all. Data are necessary.

Many thanks for this comment. The percentage of pig production costs attributed to antimicrobial use have been added in the abstract (line 29).

KEYWORDS
The following keywords are not adequate: AMR, Vietnam and behaviours.

Many thanks for your comment. In response, ‘behaviours’ has been changed to ‘antimicrobial use behaviours’ so that this is more specific. In addition to ‘AMR’ the keyword ‘antimicrobial resistance’ has been added and ‘Vietnam’ has been removed (lines 33-34)

INTRODUCTION
Line 71. What is GDP?

This has now been defined as Gross Domestic Product (line 71).

Lines 86-88. Objectives are not adequately expounded.

Many thanks for your comment. In response the authors have added additional information to outline the importance of the data collected in the study (lines 97-90). Additional information has also been added on the context of the study in a larger research project (line 93-94). Finally, the relevance of the study results for future policy development have also been added to the manuscript as an objective (lines 95-96).

RESULTS
Line 92. More information about farms is missed. Please, give standard deviation; at least in
table 1.

Many thanks for your comment. The farm data is not normally distributed therefore it is the median that is presented. The ‘average number of pigs at the time of survey’ has been replaced with ‘median number of pigs at the time of survey’ so that this is clear. In response to the comment the interquartile range has therefore been added in place of the standard deviation to table 1.

Table 1. Is “number of farm workers” considering all the farms?

The ‘number of farm workers’ is the total number of workers across all of the farms for which the information below 1 refers to in table 1. The title in table 1 has been updated from ‘number of farm workers’ to ‘number of farm workers across the farms’.

Line 113. Routine antimicrobial use. Does “use” refer to doses, animal or amount of
antimicrobial active product? It is not clear.

This refers to the act of any antimicrobial being administered as a routine to the pigs on the farms. In response the authors have changed ‘routine antimicrobial use’ to ‘the routine use of antimicrobial products’ in order to ensure that this is clear (line 124).

Lines 161-166. This not enough to study “Farmer perceptions on the economics of disease and
antimicrobial use”.

This results in lines 161 to 166 are not supposed to be a detailed economic evaluation of antimicrobial use but explore farmer perceptions of the economic importance of antimicrobials to their farms. The majority of farmers (72%) perceived that there was an economic advantage to antimicrobial use. This in addition to the economic analysis, which showed the low costs of antimicrobials in comparison to other production costs, identifies barriers to changing behaviours. Understanding barriers to changing farmer antimicrobial use behaviour is essential if governments are to introduce policy which is successful in reducing antimicrobial use in livestock. This type of question is commonly included in Knowledge, Attitudes and Practices (KAP) studies which are commonly used by the FAO to gauge human behaviours around antimicrobial use.

Table 2. Delete it, because this is for a journal and no a newspaper.             

The authors disagree as this table supports the evidence base for policy makers on antimicrobial use behaviours as detailed above. These are qualitative data and offer more detailed insights into antimicrobial use behaviours than is possible through purely quantitative data collection methods. Therefore, these data support the quantitative economic data analysis also presented.

Table 3. These results are very light.

Unfortunately collection of detailed economic data is not a common practice in the small to medium commercial pig sector in Vietnam. Therefore, the results presented are what was able to be collected on the farms as part of the study.

Lines 180-185. These are material and methods.

This paragraph has now been moved to the materials and methods section under economic analysis section (lines 390-395).

Lines 202-209. These are material and methods.

Many thanks for the comment. An additional line has been added to the methods to explain that the gross margin analysis is not seasonal (lines 395-396). However, the other information in this section gives essential background information to the table presented (table 4). Therefore the authors believe that it is easier to interpret the results with this information remaining within the results.

DISCUSSION
Lines 233-247. This is stuffing text. Please, reduce it or delete it.

Many thanks for this comment. This section has now been reduced in length (lines 244-258).

Lines 311-313. These are good conclusions.

Many thanks for your positive feedback.

MATERIALS AND METHODS
Lines 322-323. These are objectives; it is not methodology.

Many thanks for this observation. On reflection this sentence is not required as it is repeating the information given in the title, introduction and results (lines 233-235).

Lines 323. It says “small-medium commercial pig sector”. How are the farms? Number and kind
of animals?

This statement has now been deleted as above. However, an additional statement has been added to provide information on the size of the farms included in the sample with small farms having <19 breeding sows and medium farms having 20-49 breeding sows. There were no definitions for the number of fattening pigs as these were considered to be proportional to breeding sows on farrow to finish farms (lines 343-344).

Lines 360-363. The economics of antimicrobial use is not compared. There is not information
beyond what is spent; who do you know the antimicrobial effect?

This outlines the data that were collected as part of the questionnaire study. However, as outlined in the paper there were knowledge gaps in the economic data as many of the farms did not routinely collect economic or productivity data. The authors were able to demonstrate the overall low contribution of antimicrobials to overall pig production costs. Thus, this suggests antimicrobial costs are not a major consideration in farmer antimicrobial use decisions (lines 229-231).

CONCLUSIONS
The conclusions are very weak; these were well known statements.

Many thanks for your comment. There is still relatively limited research work published on the use of antimicrobials in livestock in Vietnam with studies on the subject of antimicrobial use in pigs even more scarce. Therefore, whilst some of these conclusions may be true of other countries, or indeed antimicrobial use in general, these conclusions are specific to Vietnam and provide concluding statements to the proceeding discussion.

Reviewer 3 Report

This manuscript by Coyne et al. outlines a recent study of antimicrobial use and economics in small scale pig farms in Vietnam. They compared farms in two provinces, by conducting face to face interviews, with follow-up visits and data collection by farmers over a 6-week period. They found that routine antimicrobial use was high, with most being administered via injection, followed by in-feed formulations. In addition, they found many feed products did not clearly state ingredients, therefore it is likely that antimicrobial use in feed is under-reported. They also found that 20% of antimicrobials used belonged to WGO HP-CIA category. Most farmers felt the government and veterinarians were responsible for antimicrobial stewardship, and not farmers themselves. The authors have nicely highlighted the tight economic margins faced by pig farmers in Vietnam, and how disease of animals can impact upon farmer’s livelihood.

This article is a very well written and interesting study. It highlights a very important problem which many farmers around the world face, that of balancing economics with antimicrobial stewardship. I have only very minor comments for the authors to address:

Line 67 – Bacterial names should be italicized, “bacteria to” could be removed from this sentence.

Line 133 – The authors state 83%  of HP-CIA was injectable, I think it would be good to clarify if the remaining 17% were feed additives. (I presume this is the case?)

Line 231 – should read “logistically difficult in low and middle….” Or “logistically difficult for low and middle….”

Line 287 – I believe the sentence starting “Evidence….” May be missing a word?

The authors explain in the methods that the farms had previously been enrolled in a study around antimicrobial use the year before this study. I think the authors should state this in the results as well, as this could potentially impact farmers perceptions of antimicrobials/stewardship, and therefore good to bring to the reader’s attention early on.

Author Response

The authors would like to thank the reviewer for such positive and valuable comments. The minor comments have been addressed below with the response shown in italics. The authors greatly appreciate the time the reviewer has given to this process.

This manuscript by Coyne et al. outlines a recent study of antimicrobial use and economics in small scale pig farms in Vietnam. They compared farms in two provinces, by conducting face to face interviews, with follow-up visits and data collection by farmers over a 6-week period. They found that routine antimicrobial use was high, with most being administered via injection, followed by in-feed formulations. In addition, they found many feed products did not clearly state ingredients, therefore it is likely that antimicrobial use in feed is under-reported. They also found that 20% of antimicrobials used belonged to WGO HP-CIA category. Most farmers felt the government and veterinarians were responsible for antimicrobial stewardship, and not farmers themselves. The authors have nicely highlighted the tight economic margins faced by pig farmers in Vietnam, and how disease of animals can impact upon farmer’s livelihood.

This article is a very well written and interesting study. It highlights a very important problem which many farmers around the world face, that of balancing economics with antimicrobial stewardship. I have only very minor comments for the authors to address:

Many thanks for such positive comments on the manuscript overall and for identifying the minor comments outlined below which will enhance the quality of the manuscript.

Line 67 – Bacterial names should be italicized, “bacteria to” could be removed from this sentence.

Many thanks for this observation the bacterial names have been italicized (lines 69 and 70).

Line 133 – The authors state 83%  of HP-CIA was injectable, I think it would be good to clarify if the remaining 17% were feed additives. (I presume this is the case?)

Many thanks for this comment. The remaining 17% were in fact in an in-feed formulation and this has now been added to the manuscript (line 144).

Line 231 – should read “logistically difficult in low and middle….” Or “logistically difficult for low and middle….”

 The ‘in’ has now been removed from this sentence (line 242).

Line 287 – I believe the sentence starting “Evidence….” May be missing a word?

 Many thanks for this observation the word ‘from’ has been added after ‘evidence’ and the ‘highlight’ has been made plural.

The authors explain in the methods that the farms had previously been enrolled in a study around antimicrobial use the year before this study. I think the authors should state this in the results as well, as this could potentially impact farmers perceptions of antimicrobials/stewardship, and therefore good to bring to the reader’s attention early on.

Many thanks for this observation. In response, a sentence to give context and background to the study farms has been added in the results (line 298).

Round 2

Reviewer 2 Report

The authors have put in a lot of effort and improved the manuscript.